# Timely Hepatitis C RNA Testing and Treatment in the Era of Direct-Acting Antiviral Therapy among People with Hepatitis C in New South Wales, Australia

**DOI:** 10.3390/v14071496

**Published:** 2022-07-08

**Authors:** Mohammad T. Yousafzai, Maryam Alavi, Heather Valerio, Behzad Hajarizadeh, Jason Grebely, Gregory J. Dore

**Affiliations:** The Kirby Institute, UNSW Sydney, Sydney 2052, Australia; msalehialavi@kirby.unsw.edu.au (M.A.); hvalerio@kirby.unsw.edu.au (H.V.); bhajarizadeh@kirby.unsw.edu.au (B.H.); jgrebely@kirby.unsw.edu.au (J.G.); gdore@kirby.unsw.edu.au (G.J.D.)

**Keywords:** hepatitis C virus, direct acting antiviral therapy, HCV RNA testing, treatment uptake, data linkage

## Abstract

This study aimed to identify the factors associated with timely (within four weeks) HCV RNA testing and timely (within six months) DAA initiation following HCV notification in the DAA era. We conducted a cohort study of people with an HCV notification in NSW, Australia. Notifications of positive HCV serology were linked to administrative datasets. Weights were applied to account for spontaneous clearance. Logistic regression analyses were performed. Among 5582 people with an HCV notification during 2016–2017, 3867 (69%) were tested for HCV RNA, including 2770 (50%) who received timely testing. Among an estimated 3925 people with chronic HCV infection, 2372 (60%) initiated DAA therapy, including 1370 (35%) who received timely treatment. Factors associated with timely HCV RNA testing included age (≥30 years), female sex, non-Aboriginal ethnicity, country of birth being Australia, and no history of drug dependence. Factors associated with timely treatment were age (≥30 years), male sex, non-Aboriginal ethnicity, country of birth being Australia, no history of drug dependence, and HCV/HIV co-infection. In the DAA era, 50% of people with an HCV notification did not receive timely HCV RNA testing. Most people with an HCV infection received therapy; however, DAA initiation was delayed among many.

## 1. Introduction

Globally, 58 million people live with hepatitis C virus infection (HCV), with an estimated 1.5 million new infections and 290,000 related deaths annually [1]. Chronic HCV infection is mostly asymptomatic; however, 20–30% may develop liver complications if untreated [2,3]. The global viral hepatitis strategy endorsed by World Health Assembly in 2016, set the goals of HCV elimination by 2030, including a 90% diagnosis and an 80% coverage of antiviral therapy among people with HCV infection [4]. With the availability of highly effective direct-acting antiviral (DAA) therapy, achieving this ambitious goal is feasible; however, barriers at each stage of the care cascade from HCV screening to RNA testing, and linkage to care for treatment initiation need to be addressed by contextual interventions and public health policies [5].

The timely diagnosis of HCV and treatment initiation is critical for the prevention of complications and ongoing transmission. In the DAA era, barriers to HCV care such as treatment toxicity and a longer duration of treatment have been removed but testing and treatment remains suboptimal [6,7,8]. In 2016, the Australian government made available unrestricted access to DAA for treatment of chronic HCV infection [9]. Approximately 33,000 people were treated in 2016, followed by a steady decline in more recent years [10,11]. Using well-established data linkage mechanisms in New South Wales (NSW), Australia, we aimed to evaluate time to HCV RNA testing, treatment initiation, and factors associated with timely RNA testing and DAA treatment initiation after HCV notification during the DAA era.

## 2. Materials and Methods

### 2.1. Study Design and Setting

This was a retrospective cohort study using the NSW linked data. The study setting was NSW, Australia. Nationally, NSW accounts for 35% of HCV infections and 40% of people who inject drugs (PWID) [12,13].

### 2.2. Data Sources and Record Linkages

NSW Notifiable Conditions Information Management System (NCIMS), which holds the records of all HCV- and hepatitis B virus-positive serology results (notifications) was probabilistically and deterministically linked to several administrative databases, using full name or 2 × 2 name codes (only for linkage to HIV diagnosis data), sex, date of birth, and address. HCV notifications were first internally linked within the NCIMS to identify people with HCV/HBV co-infection and then linked to: (1) inpatient hospitalization discharges (NSW Admitted Patient Data Collection); (2) deaths (NSW Registry of Birth Deaths, and Marriages); (3) opioid agonist therapy (OAT) authority (NSW Electronic Recording and Reporting of Controlled Drugs system); (4) incarcerations (NSW Bureau of Crime Statistics and Research); (5) HIV diagnosis (National HIV Database); and (6) NSW Perinatal Data Collection (PDC). A second round of linkage was performed between HCV notifications and the Pharmaceutical Benefits Scheme (PBS) and Medicare Benefits Schedule (MBS) databases using the same set of identifying information [14]. The PBS database contains information on prescription medicines that qualify for a benefit under the national health act 1953, and MBS database contain information about services, especially laboratory investigations, that qualify for a benefit under the health insurance act 1973 [15,16]. Record linkages were undertaken by the New South Wales Centre for Health Record Linkage and the Australian Institute of Health and Welfare Data Integration Services Centre.

### 2.3. Study Period

Data were extracted from each database as follows: HCV and HBV notifications (1993–2017); hospitalizations (2001–30 June 2018); deaths (1993–30 June 2018); OAT authority (1985–19 September 2018); incarcerations (1994–2017); HIV diagnosis (1985–2017); birth events (1994–2016); PBS (2010–2018); and MBS (2010–2018).

### 2.4. Study Population

The study population included people who had a first record of HCV notification in NSW, Australia, during 2016–2017. The vast majority of HCV notifications are based on positive serology (HCV antibody) rather than virology (HCV RNA); therefore, importance weights were applied to individual records to account for sex-specific rates of spontaneous clearance to estimate population with chronic HCV infection [14]. Weights were calculated based on the DAA treatment status and rates of spontaneous clearance using the formula given below [14]. Spontaneous clearance rates of 25% among men and 34% among women were selected based on the published studies from linkage data with high HCV RNA testing [17]. Treated population were weighted 1 and untreated male and female were weighted 0.65 and 0.56, respectively.
$$1.00-\frac{\mathrm{Total}\;\times \;\mathrm{spontaneous}\;\mathrm{clearance}\;\mathrm{rate}}{\mathrm{Total}-n\;\mathrm{treated}}$$

### 2.5. Exclusion Criteria

There was a total of 115,669 HCV notifications by the end of 2017. Data on HCV RNA testing and treatment initiation was available since 2010 and, hence, HCV notifications prior to 2010 (*n* = 88,285) were excluded. To allow time for treatment initiation, records were removed if death occurred within six months of HCV notification (*n* = 491). Further, post-mortem notifications (*n* = 8), duplicate notifications (*n* = 591), and those with no Medicare number (universal healthcare, required for linkage to MBS and PBS) (*n* = 2490) were excluded. The characteristics of those with and without Medicare number have been previously reported [14]. Of the remaining 23,804 HCV notifications, 18,222 were in the pre-DAA era (pre-2016) and excluded, leaving *n* = 5582 notified during 1 January 2016 to 31 December 2017 (DAA era) for this analysis (Figure 1).

### 2.6. Exposure Variables

Variables included in the analyses of factors associated with HCV RNA testing and DAA treatment comprised age (≤29, 30–44, 45–59, and ≥60 years) at HCV notification, sex (male, female), Aboriginal ethnicity, region of residence at the time of HCV notification (metropolitan, outer-metro, and rural/regional), country of birth (Australia, overseas), incarceration (no history, distant incarceration before 2016, or recent incarceration during 2016–2018), and coinfection status (HCV/HBV and HCV/HIV). Due to a small number of records with HCV/HBV/HIV coinfection (<0.5% of all included notifications), these records were classified as HCV/HIV coinfection [14]. Evidence of drug dependence was included, defined by hospital admissions due to injecting drug use and/or infections indicative of injecting drug use, or receipt of OAT [18]. The study population were considered to have no evidence of drug dependence, distant drug dependence (record occurring before 2016 only), or recent drug dependence (2016–2018). A history of alcohol use disorder (AUD) was included, defined by first time alcohol-related hospital admissions [19]. Finally, a history of end-stage liver disease (ESLD) was included and defined by first time hospital admission due to decompensated cirrhosis, bleeding esophageal varices, chronic hepatic failure (including hepatic encephalopathy), alcoholic hepatic failure, hepatorenal syndrome, and hepatocellular carcinoma [20]. Hospital admissions were coded using the International Statistical Classification of Diseases and Related Health Problems 10th Revision (ICD-10), at primary or secondary diagnosis at hospitalization [14].

### 2.7. Outcome

The main study outcomes included time from HCV notification to HCV RNA testing and time from HCV notification to DAA initiation. Timely HCV RNA testing and treatment initiation were defined to occur within four weeks and six months of HCV notification, respectively. HCV RNA testing after four weeks and treatment initiation after six months of HCV notification were defined as delayed testing and treatment initiation, respectively. Ever HCV RNA testing and DAA treatment were defined to occur anytime during the study period (2016–2018).

### 2.8. Statistical Analysis

#### 2.8.1. Analysis of Time to Testing and Treatment Initiation Using Kaplan–Meier Failure Curves

Observation time for HCV RNA testing started from date of HCV notification, and ended on date of HCV RNA testing, date of death, or 31 December of 2018, whichever occurred first. Observation time for HCV treatment started from date of HCV notification or date of 18th birthday if after and ended on the date of treatment initiation, date of death, or 31 December 2018, whichever occurred first. Time from HCV notification to HCV RNA testing was divided into four categories (<1 week, 1–4 weeks, ≥4 weeks, and never tested). Time from HCV notification to HCV treatment was divided into ≤six months, >six months, and never DAA-treated. Associations with time to HCV RNA testing and treatment initiation was assessed using Kaplan–Meier failure curves. Kaplan–Meier failure curves for estimated proportion of DAA treatment were compared among people tested for HCV RNA in <1 week, 1–4 week, ≥4 weeks, and never tested.

#### 2.8.2. Factors Associated with HCV RNA Testing and Treatment Initiation

Unadjusted and adjusted binary logistic regression models were used to evaluate factors associated with timely HCV RNA testing, ever HCV RNA testing within the study period (2016–2018), timely HCV treatment initiation, and ever DAA treatment initiation during the study period (2016–2018). Several variables were considered for inclusion in these analyses, including age at time of HCV notification, sex, Aboriginal status, country of birth, area of residence at time of HCV notification, drug dependence, HCV coinfection status, AUD, and ESLD. Variables with *p* values ≤ 0.25 at univariate level or with known clinical significance were carried into the multivariable logistic regression analysis. Variance–covariance matrices were used to test the collinearity between variables present in the adjusted models. Similarly, likelihood ratio tests were used to test the goodness of fit of the adjusted models. Since information about HCV RNA testing among the incarcerated population is not available in the MBS database, incarceration was not included in the adjusted logistic regression model for HCV RNA testing. Adjusted odds ratios (aOR) with 95% confidence intervals (95% CI) were calculated for each factor. The data were analyzed in Stata version 16.0 (College Station, TX, USA).

## 3. Results

### 3.1. Descriptive Characteristics

Among 115,669 HCV notifications at the end of 2017, 5582 were notified during the DAA era (2016–2017) and included in this analysis. A total of 3867 (69%) were tested for HCV RNA, including 2770 (72%) within four weeks of HCV notification (timely) and 1097 (28%) following four weeks (delayed) (Figure 1).

#### 3.1.1. HCV RNA Testing during the DAA Era

Key demographic characteristics of the study population are outlined in Table 1. Among 5582, more than two thirds (69%) of the population were male, 21% with Aboriginal ethnicity, 15% born outside Australia, and 44% rural/regional residents. Other characteristics included recent drug dependence (33%), recent incarceration (23%), history of AUD (8%), and history of ESLD (2%).

Among the total study population, 2770 (50%) had timely HCV RNA testing. Timely HCV RNA testing was highest among people ≥60 years (61%), and lowest among ≤29 years (34%). Timely HCV RNA testing was also lower among those with Aboriginal ethnicity (34%), and those with recent drug dependence (40%) (Table 1).

#### 3.1.2. DAA Treatment Initiation

After applying importance weights, the estimated chronic HCV infection (eligible for treatment during 2016–2018) was 3925. Among this population, 2372 (60%) were treated with DAA therapy: 1370 (35%) within six months (timely) and 1002 (26%) following six months (delayed).

Timely DAA treatment initiation by age was highest among people ≥60 years (48%) and lowest among those less than 30 years (24%). Timely DAA treatment initiation was also lower among females (29%), Aboriginal ethnicity (22%), HCV/HBV co-infected (20%), those with recent incarceration (27%), recent drug dependence (27%), and a history of AUD (32%). Timely DAA treatment initiation was higher among those who received HCV RNA testing within one week (48%) or 1–4 weeks (46%) compared with more than 4 weeks of HCV notification (27%) (Table 2). Among males, HCV RNA testing and DAA treatment initiation was similar (67% and 65%, respectively). In contrast, among females, HCV RNA testing was higher than DAA treatment initiation (75% and 51%, respectively) (Figure 2).

#### 3.1.3. Factors Associated with Timely HCV RNA Testing

The adjusted odds of timely HCV RNA testing were highest among people aged ≥60 years as compared to ≤29 years (aOR: 1.97; 95% CI 1.60, 2.43) followed by 45–59 years (aOR: 1.90; 95% CI 1.60, 2.24) and 30–44 years (aOR: 1.52; 95% CI 1.29, 1.77). Timely HCV RNA testing was also associated with females (aOR: 1.26; 95% CI 1.12, 1.42), and HCV/HIV co-infection (aOR: 2.83; 95% CI 1.68, 4.77). Timely HCV RNA testing was less likely among the Aboriginal ethnicity (aOR: 0.57; 95% CI 0.49, 0.66), those born outside Australia (aOR: 0.82; 95% CI 0.70, 0.97), and those with recent drug dependence (aOR: 0.63; 95% CI 0.55, 0.72) (Table 3). Factors associated with ever testing for HCV RNA during the study period (2016–2018) were similar to timely HCV RNA testing, including age, sex, co-infection status, area of residence, Aboriginal ethnicity, and drug dependence (Table 3).

#### 3.1.4. Factors Associated with Timely and Ever DAA Treatment Initiation

The adjusted odds of timely treatment initiation were highest among people aged ≥60 years (aOR: 2.14; 95% CI 1.64, 2.79) followed by 45–59 years (aOR: 1.71; 95% CI 1.36, 2.13), and 30–44 years (aOR 1.31; 95% CI 1.07, 1.61) as compared to ≤29 years. Timely treatment initiation odds were also higher among HCV/HIV co-infected (aOR: 1.78; 95% CI 1.07, 2.99), but lower among HCV/HBV co-infected (aOR: 0.49; 95% CI 0.26, 0.91). Timely treatment initiation odds were lower among females (aOR: 0.64; 95% CI 0.54, 0.75), those with Aboriginal ethnicity (aOR: 0.59; 95% CI 0.48, 0.73), those born outside Australia (aOR: 0.69; 95% CI 0.56, 0.85), and recent drug dependence (aOR: 0.65; 95% CI 0.55, 0.77) (Table 4).

Factors associated with ever DAA treatment initiation within the study period (2016–2018) were similar to timely treatment initiation, including age, sex, country of birth, Aboriginal ethnicity, co-infection status, and area of residence (Table 4). Although there was no association between the timing of HCV RNA testing following notification and ever HCV treated within the study period, those with delayed testing did have the slower initial treatment uptake (Figure 3).

## 4. Discussion

The majority of people recently notified with HCV have received confirmatory HCV RNA testing and initiated DAA therapy, but gaps remain in the HCV care cascade. Among people who have been notified with HCV during the DAA era (2016–2017), around a third of people had no record of HCV RNA testing during the study period. Factors associated with non-testing or delayed HCV RNA testing included a younger age, the male gender, Aboriginal ethnicity, born outside Australia, and recent drug dependence. An encouraging 60% of people with HCV notification and chronic HCV received DAA treatment, but this leaves a sizeable proportion to be linked to treatment. Factors associated with no or delayed treatment were similar with those for HCV RNA testing, although female gender and HBV/HCV coinfection were noteworthy associations. This study highlights important sub-populations with gaps in timely HCV testing and treatment initiation where interventions should be implemented to facilitate HCV elimination efforts, thereby informing health service delivery and policy.

Although the absence of HCV RNA testing was clearly associated with a lack of HCV treatment initiation, when performed, the timing of HCV RNA testing did not influence cumulative treatment uptake. People with delayed testing initially showed slower treatment uptake, but 2–3 years following notification there was similar treatment uptake. More innovative HCV screening strategies could still improve both time to HCV RNA testing and treatment uptake. Multiple clinic visits for assessment and testing, and the delay in availability of test results are considered as major barriers in the HCV care continuum, resulting in attrition at each subsequent visit [21]. Studies from the high-income countries have reported that 46% to 73% of people with HCV antibodies receive RNA testing, [22,23] and within low and middle-income countries, this proportion may be lower. State-of-the-art diagnostic platforms and simplified algorithms, such as HCV RNA reflex testing, point-of-care testing for both HCV antibody, and HCV RNA, should provide enhanced timeliness, including potential single-visit screening and linkage to care and treatment [23,24,25].

Factors associated with both timely HCV RNA testing and treatment initiation included recent drug dependence, gender, and age. What was somewhat surprising was that females had more timely HCV RNA testing, but less timely HCV treatment initiation. An older age was associated with both timely HCV RNA testing and HCV treatment initiation, possibly related to a more advanced disease. Although those with evidence of recent drug dependence had less timely HCV RNA testing and HCV treatment initiation, there was no association with having ever received treatment within the study period (2016–2018). This relative equity of HCV treatment access is encouraging and evidence that diverse models of care, including those embedded within harm reduction services, have been successful in providing linkage to HCV care. In contrast, most settings have a poorer HCV treatment initiation for highly marginalized populations. A population-based study in British Columbia, Canada, reported associations between a younger age (born after 1974), current or past injecting drug use, and poor RNA/genotype testing with treatment initiation [17].

Our finding of less timely HCV treatment initiation among women is concerning. We have previously demonstrated that women of childbearing age were less likely to have initiated treatment than men [18,26]. A global systematic review reported women who use drugs experience multiple stigmas related to societal expectations of womanhood, stereotypes of promiscuity, drug use stigma for women in healthcare, and gender-based violence contributing to a higher vulnerability and disengagement with health services [27]. Pregnancy and childbirth could also disproportionately affect younger women to delay treatment initiation. Strategies that aim to reduce vulnerabilities incurred among women, including adequate antenatal HCV screening and improved counselling and support for younger women with HCV, are required.

The area of residence at the time of HCV notification had a relatively limited impact on timely HCV RNA testing or HCV treatment initiation. In fact, a rural/regional residence was associated with having ever received HCV treatment during the study period (2016–2018). Again, this likely demonstrates the equity of HCV treatment access through a diverse range of models of care, including the involvement of primary care physicians in all areas. Urban socioeconomic factors may play a role in a somewhat lower HCV treatment initiation, consistent with a population-based study from British Columbia, Canada, that reported an association between residence in materially deprived areas and poor HCV RNA testing and treatment initiation [17].

The strongest association with HCV RNA testing and treatment initiation was observed with HCV/HIV co-infection, with a timelier HCV RNA testing and treatment. In contrast, HCV/HBV coinfection decreased timely treatment initiation. Australia has particularly high levels of HIV anti-retroviral therapy, including that delivered through primary care, providing the ideal platform for enhanced HCV screening and treatment [28]. Of concern, despite guidance to prioritize treatment of people with liver disease progression-related cofactors, HCV treatment among the HBV co-infected population was delayed. Early DAA era reports of HBV decompensation in HCV/HBV co-infected patients receiving HCV therapy [29] may have influenced therapy decision-making. The further education of both the patient and clinician is required to ensure a high HCV treatment uptake among those with HCV/HBV coinfection.

Other factors associated with timely HCV RNA testing and HCV treatment were Aboriginal ethnicity and a country of birth outside Australia. A lower HCV treatment initiation among those with Aboriginal ethnicity and among those born outside of Australia is concerning, particularly given evidence of higher ongoing infection rates in the former population [28]. The provision of culturally appropriate care, interpreters, stable housing, trained Aboriginal and non-Australian born healthcare workers as treatment advocates/community mobilizers, and reducing stigma in the primary healthcare settings, are some of the interventions to build positive, enabling relationships and enhance the HCV care cascade among those with Aboriginal ethnicity and those born outside of Australia [30].

There are several limitations that need to be considered while interpreting the findings of this study. Firstly, as HCV notifications are largely based on HCV antibody detection and MBS data do not provide HCV RNA test results, we adjusted HCV notifications for spontaneous clearance by applying sex-specific weights to estimate treatment-eligible people with HCV infection. Given that our estimates of spontaneous clearance were conservative, HCV treatment uptake could be higher. This could partly explain the lower HCV treatment uptake among the female population in this study. Secondly, HCV RNA testing information was not available for the incarcerated population and, hence, incarceration as a potential predictor of timely HCV RNA testing could not be evaluated in the adjusted model.

## 5. Conclusions

In the DAA era, about half of people with an HCV notification did not receive timely HCV RNA testing. Most people with an HCV infection received therapy; however, DAA initiation was delayed among many. Several demographic factors predict a poorer cascade of care. Enhanced strategies are required for a younger population, women, those with Aboriginal ethnicity, born outside of Australia, and those with an HCV/HBV co-infection. Innovative interventions to enhance a timely diagnosis and treatment uptake such as point-of-care screening technologies, simplified treatment algorithms, and a population-appropriate model of care are required.

## Figures and Tables

**Figure 1 viruses-14-01496-f001:**
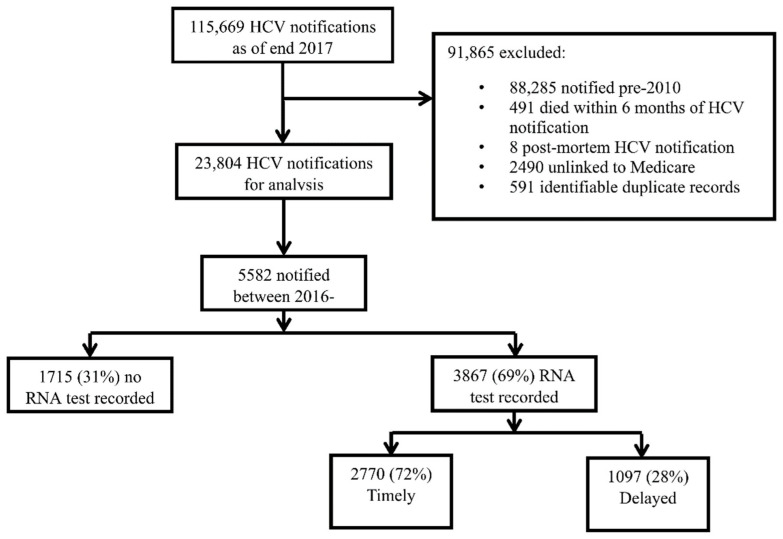
Participant disposition among people with an HCV notification in NSW 1993–2018, N = 115,669.

**Figure 2 viruses-14-01496-f002:**
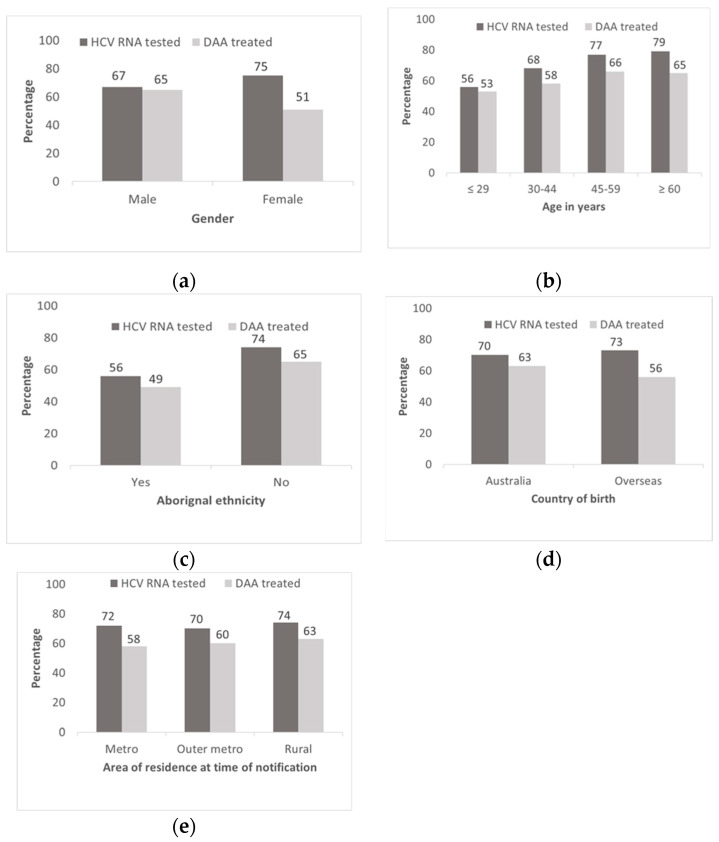
Proportion of HCV RNA testing and DAA treatment by key factors among people with an HCV notification in NSW 2016–2018; (**a**) gender; (**b**) age groups; (**c**) Aboriginal status; (**d**) country of birth; (**e**) area of residence.

**Figure 3 viruses-14-01496-f003:**
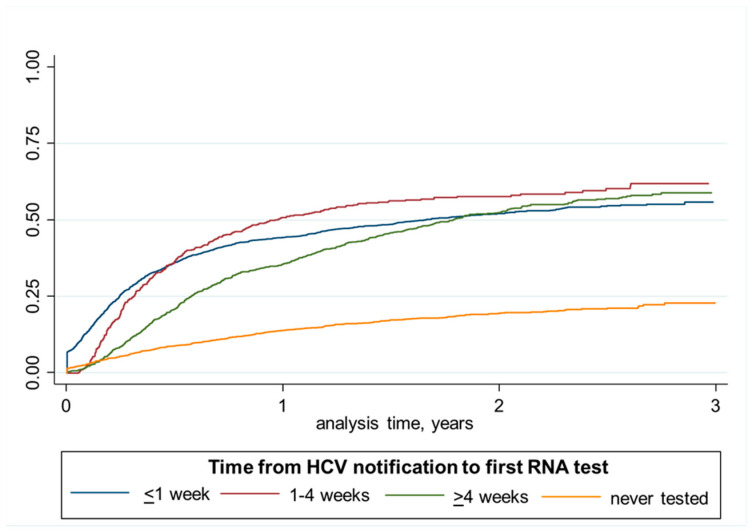
Kaplan–Meier failure curve comparing the proportion of DAA treatment across different times to first RNA testing among people with an HCV notification in NSW 2016–2018.

**Table 1 viruses-14-01496-t001:** The distribution of sociodemographic and clinical characteristics of people with an HCV notification across different periods of time to RNA testing in NSW 2016–2018 (N = 5582).

		Total	RNA Tested			Never RNA Tested *n* (%) **
		N (%) ^§^	Ever Tested *n* (%) **	Tested < 4 Weeks *n* (%) **	Tested > 4 Weeks *n* (%) **	
**Total**		5582 (100)	3867 (69)	2770 (50)	1097 (20)	1715 (31)
**Age at HCV Diagnosis ***	≤29 years	1203 (22)	674 (56)	411 (34)	236 (20)	529 (44)
	30–44	2057 (37)	1398 (68)	981 (48)	417 (20)	659 (32)
	45–59	1637 (29)	1252 (77)	959 (59)	293 (18)	385 (24)
	≥60	684 (12)	542 (79)	418 (61)	124 (18)	142 (21)
**Sex ***	Male	3862 (69)	2572 (67)	1838 (48)	734 (19)	1290 (33)
	Female	1717 (31)	1293 (75)	930 (54)	363 (21)	424 (25)
**Aboriginal Ethnicity ***	No	3625 (65)	2680 (74)	1942 (54)	738 (20)	945 (26)
	Yes	1180 (21)	658 (56)	396 (34)	262 (22)	522 (44)
**Country of Birth ***	Australia	4042 (72)	2819 (70)	1972 (49)	847 (21)	1233 (30)
	Overseas	853 (15)	625 (73)	453 (53)	172 (20)	228 (27)
**Co-Infection Status**	HCV only	5391 (97)	3725 (69)	2658 (49)	1067 (20)	1666 (31)
	HCV/HBV	108 (2)	65 (60)	49 (45)	16 (15)	43 (40)
	HCV/HIV	83 (2)	77 (93)	63 (76)	14 (17)	6 (7)
**Area of Residence at The Time of HCV ***	Metro	1100 (20)	796 (72)	606 (55)	190 (17)	301 (28)
	Outer Metro	1533 (28)	1071 (70)	762 (50)	309 (20)	462 (30)
	Rural/regional	2469 (44)	1834 (74)	1324 (54)	510 (21)	635 (26)
**Incarcerated**	No history	3473 (62)	2685 (77)	2060 (59)	625 (18)	788 (23)
	Distant	836 (15)	551 (66)	361 (43)	190 (23)	285 (34)
	Recent	1273 (23)	631 (50)	349 (27)	282 (22)	642 (50)
**Drug Dependence**	No history	3077 (55)	2206 (72)	1703 (55)	503 (16)	871 (28)
	Distant	656 (12)	453 (69)	329 (50)	124 (19)	203 (31)
	Recent	1849 (33)	1208 (65)	738 (40)	470 (25)	641 (35)
**History of AUD**	No history	5137 (92)	3549 (69)	2566 (50)	983 (19)	1588 (31)
	History	445 (8)	318 (72)	204 (46)	114 (26)	127 (29)
**History of ESLD**	No history	5452 (98)	3763 (69)	2695 (49)	1068 (20)	1689 (31)
	History	130 (2)	104 (80)	75 (58)	29 (22)	26 (20)

* Missing data not shown, AUD = alcohol use disorder, ESLD = end-stage liver disease; § column percentage, ** row percentage.

**Table 2 viruses-14-01496-t002:** Distribution of sociodemographic and clinical characteristics of people with an HCV notification across different periods of time to treatment initiation in NSW 2016–2018 (N = 3925).

Characteristics		Total ^⸙^	DAA Treatment Initiation			Never DAA Treated *n* (%) **
		N (%) ^§^	Ever DAA Treated *n* (%) **	DAA Initiation ≤ 6 Months *n* (%) **	DAA Initiation > 6 Months *n* (%) **	
**Total**		3925 (100)	2372 (60)	1370 (35)	1002 (26)	1553 (40)
**Age at HCV Diagnosis ***	≤ 29	802 (20)	426 (53)	191 (24)	235 (29)	376 (47)
	30–44	1424 (36)	828 (58)	448 (32)	380 (27)	596 (42)
	45–59	1199 (31)	792 (66)	494 (41)	298 (25)	407 (34)
	≥ 60	499 (13)	326 (65)	237 (48)	89 (18)	173 (35)
**Sex**	Male	2729 (70)	1763 (65)	1029 (38)	734 (27)	966 (35)
	Female	1196 (31)	609 (51)	341 (29)	268 (22)	587 (49)
**Aboriginal Ethnicity ***	No	2633 (67)	1702 (65)	1007 (38)	695 (26)	931 (35)
	Yes	766 (20)	376 (49)	170 (22)	206 (27)	390 (51)
**Country of Birth ***	Australia	2890 (74)	1809 (63)	1016 (35)	793 (27)	1081 (37)
	Overseas	582 (15)	328 (56)	199 (34)	129 (22)	254 (44)
**Co-Infection Status**	HCV only	3792 (97)	2292 (60)	1319 (35)	973 (26)	1500 (40)
	HCV/HBV	66 (2)	28 (42)	13 (20)	15 (23)	38 (58)
	HCV/HIV	66 (2)	52 (79)	38 (58)	14 (21)	14 (21)
**Area of Residence at The Time of HCV ***	Metro	759 (19)	439 (58)	304 (40)	135 (18)	320 (42)
	Outer Metro	1079 (28)	649 (60)	356 (33)	293 (27)	430 (40)
	Rural/regional	1775 (45)	1121 (63)	633 (36)	488 (28)	654 (37)
**Time to RNA Test**	< 1 week	1689 (43)	1160 (69)	806 (48)	354 (21)	529 (31)
	1–4 weeks	411 (11)	304 (74)	187 (46)	117 (29)	107 (26)
	> 4 weeks	834 (21)	583 (70)	227 (27)	356 (43)	251 (30)
	No test recorded	991 (25)	325 (33)	150 (15)	175 (18)	666 (67)
**Incarcerated**	No history	2463 (63)	1494 (61)	948 (39)	546 (22)	969 (39)
	Distant	580 (15)	348 (60)	182 (31)	166 (29)	232 (40)
	Recent	882 (23)	530 (60)	240 (27)	290 (33)	352 (40)
**Drug Dependence**	No history	2170 (55)	1316 (61)	850 (39)	466 (22)	854 (39)
	Distant	473 (12)	303 (64)	178 (38)	125 (26)	170 (36)
	Recent	1282 (33)	753 (59)	342 (27)	411 (32)	529 (41)
**History of AUD**	No history	3598 (92)	2152 (60)	1266 (35)	886 (25)	1446 (40)
	History	327 (8)	220 (67)	104 (32)	116 (36)	107 (33)
**History of ESLD**	No history	3834 (98)	2318 (61)	1340 (35)	978 (26)	1516 (40)
	History	91 (2)	54 (59)	30 (33)	24 (26)	37 (41)

* Missing data not shown, AUD = alcohol use disorder, ESLD = end-stage liver disease, ^⸙^ Total HCV notifications weighted to adjust for the sex-specific spontaneous viral clearance, ^§^ column percentage, ** row percentage.

**Table 3 viruses-14-01496-t003:** Univariate and multivariate logistic regression analysis showing the predictors of timely (within 4 weeks of HCV notification) and ever RNA tested among people with an HCV notification in NSW 2016–2018 (N = 5582).

Characteristics, *n* (%)		RNA Tested within 4 Weeks		Ever RNA Tested	
		OR (95% CI)	aOR (95% CI)	OR (95% CI)	aOR (95% CI)
**Age at HCV Diagnosis**	≤29	reference	reference	reference	reference
	30–44	1.75 (1.52, 2.04)	1.52 (1.29, 1.77)	1.66 (1.44, 1.93)	1.40 (1.20, 1.64)
	45–59	2.72 (2.34, 3.18)	1.90 (1.60, 2.24)	2.55 (2.17, 3.00)	1.8 (1.51, 2.15)
	≥60	3.03 (2.49, 3.68)	1.97 (1.60, 2.43)	3.0 (2.41, 3.72)	2.07 (1.63, 2.62)
**Sex**	Male	reference	reference	reference	reference
	Female	1.3 (1.16, 1.45)	1.26 (1.12, 1.42)	1.53 (1.35, 1.74)	1.49 (1.30, 1.71)
**Aboriginal Ethnicity**	No	reference	reference	reference	reference
	Yes	0.44 (0.38, 0.50)	0.57 (0.49, 0.66)	0.44 (0.39, 0.51)	0.56 (0.48, 0.65)
**Country of Birth**	Australia	reference	reference	reference	reference
	Overseas	1.19 (1.03, 1.37)	0.82 (0.70, 0.97)	1.19 (1.01, 1.40)	0.85 (0.70, 1.02)
**Co-Infection Status**	HCV only	reference	reference	reference	reference
	HCV/HBV	0.85 (0.58, 1.25)	0.84 (0.56, 1.25)	0.68 (0.46, 1.00)	0.66 (0.44, 0.99)
	HCV/HIV	3.23 (1.95, 5.37)	2.83 (1.68, 4.77)	5.74 (2.50, 13.20)	5.38 (2.31, 12.51)
**Area of Residence at The Time of HCV**	Metro	reference	reference	reference	reference
	Outer Metro	0.81 (0.69, 0.94)	0.88 (0.75, 1.04)	0.89 (0.75, 1.06)	0.95 (0.80, 1.13)
	Rural/regional	0.94 (0.82, 1.09)	1.10 (0.94, 1.28)	1.10 (0.94, 1.29)	1.24 (1.04, 1.47)
**Incarcerated**	No history	reference		reference	
	Distant	0.52 (0.45, 0.61)	**	0.57 (0.48, 0.67)	**
	Recent	0.26 (0.23, 0.30)		0.29 (0.25, 0.33)	
**Drug Dependence**	No history	reference	reference	reference	reference
	Distant	0.81 (0.69, 0.96)	0.88 (0.73, 1.06)	0.88 (0.73, 1.06)	0.89 (0.73, 1.09)
	Recent	0.54 (0.48, 0.60)	0.63 (0.55, 0.72)	0.74 (0.66, 0.84)	0.85 (0.74, 0.98)
**History of AUD**	No history	reference	reference	reference	reference
	History	0.85 (0.70, 1.03)	0.93 (0.75, 1.14)	1.12 (0.90, 1.39)	1.19 (0.94, 1.49)
**History of ESLD**	No history	reference	reference	reference	reference
	History	1.40 (0.98, 1.98)	1.13 (0.79, 1.63)	1.80 (1.16, 2.77)	1.23 (0.79, 1.93)

aOR = adjusted odds ratio, CI = confidence interval, AUD = alcohol use disorder, ESLD = end-stage liver disease, ** HCV RNA testing data not available among incarcerated population and hence excluded from the adjusted model.

**Table 4 viruses-14-01496-t004:** Univariate and multivariate logistic regression analysis showing factors associated timely (within 6 months of HCV notification) and ever DAA-treated among people with an HCV notification in NSW 2016–2018 (N = 3925).

Characteristics		DAA Initiation within 6 Months		Ever DAA Treated	
		OR (95% CI)	aOR (95% CI)	OR (95% CI)	aOR (95% CI)
**Age at HCV Diagnosis ***	≤29	reference	reference	reference	reference
	30–44	1.46 (1.21, 1.79)	1.31 (1.07, 1.61)	1.23 (1.03, 1.46)	1.12 (0.93, 1.34)
	45–59	2.24 (1.84, 2.74)	1.71 (1.36, 2.13)	1.72 (1.43, 2.06)	1.49 (1.21, 1.83)
	≥60	2.90 (2.28, 3.68)	2.14 (1.64, 2.79)	1.66 (1.32, 2.10)	1.54 (1.19, 1.99)
**Sex**	Male	reference	reference	reference	reference
	Female	0.66 (0.57, 0.76)	0.64 (0.54, 0.75)	0.57 (0.49, 0.65)	0.59 (0.51, 0.68)
**Aboriginal Ethnicity**	No	reference	reference	reference	reference
	Yes	0.46 (0.38, 0.55)	0.59 (0.48, 0.73)	0.52 (0.45, 0.62)	0.53 (0.44, 0.64)
**Country of Birth**	Australia	reference	reference	reference	reference
	Overseas	0.96 (0.79, 1.15)	0.69 (0.56, 0.85)	0.77 (0.64, 0.92)	0.66 (0.53, 0.80)
**Co-Infection Status**	HCV only	reference	reference	reference	reference
	HCV/HBV	0.46 (0.25, 0.84)	0.49 (0.26, 0.91)	0.48 (0.29, 0.78)	0.52 (0.31, 0.86)
	HCV/HIV	2.52 (1.54, 4.12)	1.78 (1.07, 2.99)	2.39 (1.32, 4.30)	2.12 (1.15, 3.89)
**Area of Residence at The Time of HCV**	Metro	reference	reference	reference	reference
	Outer Metro	0.74 (0.61, 0.89)	0.84 (0.69, 1.04)	1.10 (0.91, 1.32)	1.21 (1.00, 1.48)
	Rural/regional	0.83 (0.70, 0.99)	0.89 (0.74, 1.08)	1.25 (1.05, 1.48)	1.27 (1.05, 1.52)
**Time to RNA Test**	<1 week	reference		reference	
	1–4 weeks	0.91 (0.74,1.14)		1.30 (1.02, 1.66)	
	>4 weeks	0.41 (0.34, 0.49)	**	1.06 (0.89, 1.27)	**
	No test recorded	0.20 (0.16, 0.24)		0.22 (0.19, 0.26)	
**Incarcerated**	No history	reference	reference	reference	reference
	Distant	0.73 (0.60, 0.89)	0.88 (0.71, 1.09)	0.97 (0.81, 1.18)	0.96 (0.78, 1.18)
	Recent	0.60 (0.50, 0.71)	0.91 (0.73, 1.12)	0.98 (0.83, 1.14)	1.16 (0.95, 1.42)
**Drug Dependence**	No history	reference	reference	reference	reference
	Distant	0.94 (0.76, 1.15)	1.00 (0.80, 1.25)	1.15 (0.94, 1.41)	1.08 (0.86, 1.35)
	Recent	0.57 (0.49, 0.66)	0.65 (0.55, 0.77)	0.92 (0.80, 1.06)	0.92 (0.78, 1.08)
**History of AUD**	No history	reference	reference	reference	reference
	History	0.86 (0.67, 1.10)	0.88 (0.68, 1.14)	1.38 (1.09, 1.76)	1.26 (0.98, 1.61)
**History of ESLD**	No history	reference	reference	reference	reference
	History	0.92 (0.59, 1.42)	0.73 (0.46, 1.15)	0.96 (0.63, 1.46)	0.72 (0.46, 1.11)

aOR = adjusted odds ratio, CI = confidence interval, AUD = alcohol use disorder, ESLD = end-stage liver disease, * Missing data not shown, ** Inadequate sample size to calculate aOR.

## Data Availability

This publication involved information collected by population-based health administration registries. Data used for this research cannot be deposited on servers other than those approved by ethics committees. This publication has used highly sensitive health information through linkage of several administrative datasets. De-identified linked information has been provided to the research team under strict privacy regulations. Except in the form of conclusions drawn from the data, researchers do not have permission to disclose any data to any person other than those authorized for the research project.

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
