# Peer review of "Timely Hepatitis C RNA Testing and Treatment in the Era of Direct-Acting Antiviral Therapy among People with Hepatitis C in New South Wales, Australia"

_viruses, 2022, doi:10.3390/v14071496_

Round 1

Reviewer 1 Report

 1.      In the title, what is “Time to hepatitis C RNA testing and treatment”? Please make changes of title.

2.      Result sections seems descriptive. In the end of each section, add short summary.

3.      Use better figure 3.

Author Response

Response to reviewers
Please note that all page and paragraph numbers refer to the manuscript version with tracked changes.

Reviewer #1

Comments:

  1. In the title, what is “Time to hepatitis C RNA testing and treatment”? Please make changes of title.

Response: We have revised the title by replacing "time to hepatitis C RNA testing …" with "timely hepatitis C RNA testing …" in the revised manuscript. (Page 1, line # 1)

  1. Result sections seems descriptive. In the end of each section, add short summary.

Response: Thank you for your comment. We have provided inferential results from the adjusted models. We believe adding short summaries after descriptive results is not required, particularly given this is covered in the discussion section. (Page 4, subheading 3.1.3 & 3.1.4, line # 174-195 are the Inferential results)

  1. Use better figure 3.

Response: We have modified figure 3 by converting it into high resolution 600 dpi to make it better and more visible. (Page 7, Figure 3)

Reviewer 2 Report

I have read with great interest the study by Yousafzai et al, in which the authors provide data regarding the time taken from HCV testing to RNA testing and effectively start DAA treatment. The study is interesting and provides a good opportunity to see some characteristics pertaining to the linkage to care, when DAA for HCV treatment emerged.

Some points should be addressed by the authors.

1. The concept of applying importance weights in order to adjust for missing RNA testing should be further explained, both in the Methods and in the Discussion.

2. The percentages in the Incarcerated tab of Table 2 add up to 101%. Were there subjects that had multiple sentences served, or is it a glitch?

3. The age group is >60 years old, or does it include 60 years as well? Tables seem to include the age of 60 years old in that group, while the text refers strictly to above 60.

4. This is also a minor point that will probably be corrected during editing - the authors kept the instructions for the Results and Methods sections, at the end of the last respective paragraph. 

Author Response

Reviewer #2

I have read with great interest the study by Yousafzai et al, in which the authors provide data regarding the time taken from HCV testing to RNA testing and effectively start DAA treatment. The study is interesting and provides a good opportunity to see some characteristics pertaining to the linkage to care, when DAA for HCV treatment emerged.

Response: We appreciate this kind comment by the reviewer.

  1. The concept of applying importance weights in order to adjust for missing RNA testing should be further explained, both in the Methods and in the Discussion.

Response: We have further described and added details about the weighting of HCV notifications including the formula for calculating the weights in this study. Also, some details about the use of importance weights to estimate population with chronic HCV as a potential limitation is included in the limitation. (Page 2, subheading 2.4, line# 79 - 87, and Page 12, line # 304-306)

  1. The percentages in the Incarcerated tab of Table 2 add up to 101%. Were there subjects that had multiple sentences served, or is it a glitch?

Response: Thank you for highlighting this. The reason for incarcerated tab proportion being >100% is because of the weighting. All the weighted denominators are rounded to the nearest whole number. We have added a footnote to the table 2 to clarify this. (Page 9, line # 223-224)

  1. The age group is >60 years old, or does it include 60 years as well? Tables seem to include the age of 60 years old in that group, while the text refers strictly to above 60.

Response: Age group is 60 years and above ( 60). Thank you for identifying this error. We have made all those corrections in the text also. (Page 4, line # 159, 166, 175, and 185)

  1. This is also a minor point that will probably be corrected during editing - the authors kept the instructions for the Results and Methods sections, at the end of the last respective paragraph.

Response: Thank you for your comment. We have deleted all these instructions, inadvertently copied while using the standard template provided by the journal. (Page 4, line # 195-197, Page 13, line # 308-311,)

Reviewer 3 Report

Reviewer comments:

General comments:

The study evaluating barriers in timely HCV RNA testing and treatment initiation in NSW, Australia has important public health implication for achieving WHO HCV elimination goals in NSW. The authors used NSW notifiable conditions data and linked with several other databases to create a huge dataset for the analysis to achieve the objectives of this study. Overall, the study is robust, with sufficient methodological rigors and power. Some minor comments are provided below to further improve the manuscript before publication.

Materials and Methods: authors are advised to report the design of the study e.g., retrospective cohort design or just a cohort study design. These details can be reported under the subsection 2.2 data sources and record linkages.

Study population: authors reported weighting of the HCV notifications to adjust for the sex specific spontaneous clearance. They referred to their previous publication for the detail methods of weighted calculations which is fine however, it is better to include some essential details about the weighted calculation here in this paper as well.

Statistical analysis: the authors reported using binary logistic regression to identify factors associated with timely HCV RNA testing and treatment initiation. Further details e.g., criteria for selection of variables for the final models (variables considered as confounders, or clinical significance), methods of fitting the final models, and assessing the models’ goodness of fit etc. should also be reported.

Author Response

Reviewer #3

The study evaluating barriers in timely HCV RNA testing and treatment initiation in NSW, Australia has important public health implication for achieving WHO HCV elimination goals in NSW. The authors used NSW notifiable conditions data and linked with several other databases to create a huge dataset for the analysis to achieve the objectives of this study. Overall, the study is robust, with sufficient methodological rigors and power. Some minor comments are provided below to further improve the manuscript before publication.

Response: We highly appreciate your kind comments and encouragement.

  1. Materials and Methods: authors are advised to report the design of the study e.g., retrospective cohort design or just a cohort study design. These details can be reported under the subsection 2.2 data sources and record linkages.

Response: We have added the study design under subheading 2.1. (Page 2, line # 50-51)

  1. Study population: authors reported weighting of the HCV notifications to adjust for the sex specific spontaneous clearance. They referred to their previous publication for the detail methods of weighted calculations which is fine however, it is better to include some essential details about the weighted calculation here in this paper as well.

Response: We have further described and added details about the weighting of HCV notifications including the formula for calculating the weights in this study. Also, some details about the use of importance weights to estimate population with chronic HCV as a potential limitation is included in the limitation. (Page 2, subheading 2.4, line# 79 - 87, and Page 12, line # 304-306)

  1. Statistical analysis: the authors reported using binary logistic regression to identify factors associated with timely HCV RNA testing and treatment initiation. Further details e.g., criteria for selection of variables for the final models (variables considered as confounders, or clinical significance), methods of fitting the final models, and assessing the models’ goodness of fit etc. should also be reported.

Response: Thank you for your comment. We have added more details about the binary logistic regression analysis including selection of variables in the final model, assessment for covariance and testing of the models goodness of fit. (Page 3, line# 141-144)
